# Optimized Production of Virus-like Particles in a High-CHO-Cell-Density Transient Gene Expression System for Foot-and-Mouth Disease Vaccine Development

**DOI:** 10.3390/vaccines13060581

**Published:** 2025-05-29

**Authors:** Ana Clara Mignaqui, Alejandra Ferella, Cintia Sánchez, Matthew Stuible, Romina Scian, Jorge Filippi, Sabrina Beatriz Cardillo, Yves Durocher, Andrés Wigdorovitz

**Affiliations:** 1Instituto de Virología e Innovaciones Tecnológicas, IVIT, INCUINTA, CONICET-INTA, Hurlingham 1686, Buenos Aires, Argentina; anaclaramignaqui@gmail.com (A.C.M.); aferella@hotmail.com (A.F.); 2Instituto de Investigaciones Forestales y Agropecuarias de Bariloche, IFAB, GNV, CONICET-INTA, San Carlos de Bariloche 8400, Rio Negro, Argentina; 3Biogénesis Bagó, Garín 1619, Buenos Aires, Argentina; cintia.sanchez@biogenesisbago.com (C.S.); romina.scian@biogenesisbago.com (R.S.); jorge.filippi@biogenesisbago.com (J.F.); sabrina.cardillo@biogenesisbago.com (S.B.C.); 4Human Health Therapeutics Research Center, National Research Council Canada, Montreal, QC H4P 2R2, Canada; matthew.stuible@cnrc-nrc.gc.ca (M.S.); yves.durocher@cnrc-nrc.gc.ca (Y.D.)

**Keywords:** virus-like particles, foot-and-mouth disease virus, CHO cells

## Abstract

Background/Objectives: Foot-and-mouth disease virus (FMDV) poses a continuous threat to livestock health and agricultural economies. Current vaccines require high biosafety standards and are costly to produce. While novel vaccine technologies have been explored, most fail to meet industrial scalability, cost-efficiency, or multiserotype flexibility required for effective FMD control. This study aimed to evaluate the feasibility of using a high-cell density transient gene expression (TGE) system in CHO cells for the production of FMDV virus-like particles (VLPs) as a recombinant vaccine platform. Methods: VLP expression was optimized by adjusting cDNA and polyethyleneimine (PEI) concentrations. Expression yields were compared at 24 and 48 h post-transfection to determine optimal harvest timing. We further tested the system’s capacity to express different serotypes and chimeric constructs, incorporating VP1 sequences from various FMDV strains. Immunogenicity was evaluated in swine using VLPs from the A2001 Argentina strain as a model. Results: Optimal VLP expression was achieved at 24 h post-transfection. Chimeric constructs incorporating heterologous VP1 regions were successfully expressed. Immunized pigs developed protective antibody titers as measured by a virus neutralization test (VNT, log_10_ titer 1.43) and liquid-phase blocking ELISA (LPBE, titer 2.20) at 28 days post-vaccination (dpv). Titers remained above protective thresholds up to 60 dpv with a single dose. A booster at 28 dpv further elevated titers to levels comparable to those induced by the inactivated vaccine. Conclusions: Our results demonstrate the feasibility of using CHO cell-based TGE for producing immunogenic FMDV VLPs. This platform shows promise for scalable, cost-effective, and biosafe development of recombinant FMD vaccines.

## 1. Introduction

Foot-and-Mouth Disease (FMD) is a viral disease that affects both domestic and wild cloven-hoofed animals, including cattle, sheep, goats, and pigs, among others. It is one of the most significant diseases impacting animal farming, not only due to its high contagion rate but also because extensive measures are required to prevent virus transmission and FMD propagation. Once the disease emerges in a geographical area, it leads to substantial economic losses and has major implications in the national livestock industry and international animal trade. Control measures for the disease primarily involve regular vaccination in endemic regions and the implementation of trade restrictions.

Foot-and-mouth disease virus (FMDV), an Aphtovirus of the family Picornaviridae, is the causative agent of the disease. FMDV is a non-enveloped, single-stranded positive-sense RNA virus, and the viral particle is formed by the combination of VP1, VP2, VP3 and VP4 proteins assembled into an icosahedral capsid. There are seven different virus serotypes: O, A, C, Asia 1, South African Territories (SAT) 1, SAT2 and SAT3. For a vaccine to be effective, it must protect against all serotypes present in the region. It is estimated that FMDV circulates in approximately 77% of the global livestock population and is present in over 100 countries [1].

FMD management in endemic regions has an estimated annual cost ranging from 8.4 to 27.3 billion USD. The economic impact of eradicating the 2001 FMD outbreak in the United Kingdom was estimated in 14.5 billion USD [2] while some scenarios from a recent study suggest that an FMD outbreak in Midwestern USA could cost over 180 billion USD [3].

In endemic regions, vaccination programs are economically beneficial, with a benefit–cost ratio of 5.7. This means that the economic losses due to FMDV infection are nearly six times higher than costs of vaccination [4].

The global FMD vaccine market was estimated at 1.5 billion doses in 2016, and it is expected to reach 3.0 billion doses by 2025 [5]. Currently, the Asia–Pacific region is the primary user of FMD vaccines. China represents a market of 1300 million doses, while the rest of Asia contributes a potential market of 190 million doses, and the Middle East accounts for 124 million doses.

The currently used vaccine consists of a preparation of purified inactivated virus produced in BHK-21 cells under stringent biosafety manufacturing procedures in costly high-containment facilities. Although the inactive vaccine is effective, significant efforts have been made in recent years to develop a novel recombinant vaccine that avoids the growth of live virus and its associated high costs and risks. Most recombinant alternatives rely on the production of Virus-Like Particles (VLPs) that mimic the structure of the viral particle but lack the infective RNA. Numerous studies have demonstrated that VLPs can be produced using different recombinant strategies in heterologous expression hosts such as HEK293 cells, Sf9 cells, *Saccharomyces cerevisiae* and *Escherichia coli* [6,7,8,9,10]. Additionally, genetic strategies that use plasmids, adenovirus or RNA to introduce the genetic material necessary for VLP formation in the vaccinated animal have been investigated [11,12,13,14,15,16]. However, despite the publication of effective VLP-based vaccines at small scale, none have achieved industrialization.

Interestingly, CHO cells, which are the most common heterologous expression host in the therapeutic protein industry, have never been explored for producing FMDV VLPs for vaccine purposes. One of the key advantages of CHO cells is their ability to grow in suspension cultures in various commercially available serum-free medium. They are already used in the manufacturing of over a hundred FDA-EMA-approved biologics, including three recombinant vaccines [17].

The aim of the present work was to evaluate the feasibility of using CHO cells to produce a FMDV VLP-based vaccine. We optimized VLPs expression to achieve competitive yields and assessed the system’s versatility in producing different FMDV serotypes and chimeras. Moreover, the protective immunogenicity of the CHO-produced recombinant VLPs observed in a pig immunogenicity study was comparable to that of one of the top five FMD vaccines currently used worldwide.

## 2. Materials and Methods

### 2.1. CHO-3E7 Transfection in BCDT Media

CHO-3E7 cells were grown in BalanCD Transfectory CHO media (BCDT, Irvine Scientific, Santa Ana, CA, USA) supplemented with 4 mM glutamine. CHO-3E7 cells were seeded at 48 h prior to transfection. For routine culture, cells were diluted every 2–3 days to maintain cell densities below 3 × 10^6^/mL. Before transfection, cells were used directly at a density of 7 × 10^6^ cells/mL or diluted with fresh media to 5 × 10^6^ cells/mL, with dimethylacetamide added to a final concentration of 0.075% (*v*/*v*). Plasmid DNA and PEI were diluted separately in BCDT medium. The diluted PEI was added to the diluted DNA and incubated for 7 min at room temperature. The PEI/DNA mixture was then added to cells and returned to the incubator with shaking at 37 °C. At 24 h post-transfection (hpt), cultures were harvested for protein analysis and cell counts.

### 2.2. Plasmids

The pTT-GFP plasmid was used for protocol optimization [18]. pTT5 expression vector encoding P12A polyprotein and 3C protease from FMDV, both together and separately, ‘as described previously’, [6] were used for the expression of FMDV VLPs.

### 2.3. Production and Formulation of VLP

Recombinant protein expression was carried out in CHO-3E7 cells transiently transfected with the expression plasmid pTT5, using polyethylenimine (PEI) at a DNA:PEI ratio of 1:5. Cells were cultured in serum-free BalanCD Transfectory CHO (FUJIFILM Irvine Scientific) + 4 mM L-Glutamine, at 37 °C with 5% CO_2_ under orbital shaking conditions (125 rpm). After 24 h, cells were harvested by centrifugation at 1000× *g* for 10 min at 4 °C. The pellet was washed once with ice-cold PBS and lysed in buffer containing 50 mM Tris-HCl pH 8.0, 150 mM NaCl. Cell lysis was performed using three repeated freeze–thaw cycles at −80 °C. Then, cell debris was removed by centrifugation at 12,000× *g* for 10 min at 4 °C. The resulting supernatant, containing the crude recombinant protein, was aliquoted and stored at −80 °C until further use. To assess the stability of the Virus-Like Particles (VLPs), stored samples were analyzed at 30, 90, and 120 days post-lysis. Quantification by ELISA showed no significant variation in the total protein content over time. In parallel, analysis by sucrose gradient ultracentrifugation demonstrated that the integrity and quantity of VLPs remained unchanged, indicating that both the antigenicity and physical stability of the particles were preserved during storage at −80 °C.

### 2.4. VLP Analysis: ELISA Quantification, Western Blot, Sucrose Gradient

Western blotting was performed by resolving cell lysates on 12% SDS-PAGE gels, followed by electrotransfer of proteins onto polyvinylidene difluoride (PVDF) membranes. Membranes were blocked at 37 °C for 1 h using 3% bovine serum albumin in TBS with 0.1% Tween-20 and subsequently incubated for 1 h with in-house anti-FMDV guinea pig serum (1:500), raised against the inactivated wild-type FMDV A2001 Argentina strain. After extensive washing, membranes were incubated with horseradish peroxidase (HRP)-conjugated goat anti-guinea pig secondary antibody (1:1000, KPL). Signal detection was carried out using enhanced chemiluminescence (ECL Plus, Thermo Scientific, Waltham, MA, USA) and imaged on a GBox system (Syngene, Bangalore, India).

Quantification of VLPs was achieved through an indirect ELISA. High-binding Maxisorp plates were coated overnight at 4 °C with a rabbit polyclonal anti-FMDV A2001 Argentina serum (1:3000), diluted in carbonate-bicarbonate buffer (pH 9.6). After washing with PBS containing 0.1% Tween-20, plates were blocked for 30 min at 37 °C using 5% normal equine serum in PBS-Tween. VLP samples and serial dilutions of inactivated FMDV (used as a standard curve) were then added and incubated for 1 h at 37 °C. After additional washes, a guinea pig polyclonal anti-FMDV A2001 serum (1:3000) was applied, followed by HRP-conjugated goat anti-guinea pig antibody. Tetramethylbenzidine (TMB) was used as the substrate, and the reaction was stopped after 5 min using 12% sulfuric acid. Absorbance was measured at 450 nm using a Multiskan FC microplate reader (Thermo Scientific).

Sucrose gradient fractionation was performed by layering 1 mL each of 15%, 25%, 35%, and 45% (*w*/*v*) sucrose solutions in Ultra-Clear ultracentrifuge tubes (13 × 51 mm). Samples were gently applied on top, and gradients were centrifuged at 45,000 rpm for 2 h at 4 °C in a SW 55 Ti rotor (Beckmann, Brea, CA, USA Optima-LP X-100), with acceleration and deceleration settings of 9 and 4, respectively. After centrifugation, 0.5 mL fractions were collected and analyzed for FMDV-specific protein content by ELISA, as described above.

### 2.5. Vaccine Formulations

The vaccines used in pigs were formulated as a water-in-oil single emulsion with (i) 5 μg VLPs mixed with Commercial oil Adjuvant (CA); (ii) 25 μg VLPs mixed with CA; or (iii) 50 μg VLPs mixed with CA, in a final volume of 2 mL/dose. The commercial oil adjuvant (CA) used was provided by Biogénesis Bagó (Buenos Aires, Argentina). The formulations were prepared following the manufacturer’s indications in a proportion adjuvant/antigen of 50:40. The commercial vaccine consisted in a water-in-oil single emulsion containing A/Argentina/2001 iFMDV and was provided by Biogénesis Bagó S.A. This vaccine has been approved by SENASA (OIE guide [19]).

### 2.6. Animal Experiments and Sampling Procedure

Three groups of eight-week-old pigs received a single vaccination with each vaccine, respectively, and three other groups received two vaccinations of each vaccine, respectively, with an interval of 4 weeks. A control group received a single vaccination of a monovalent A2001 Argentina inactivated vaccine. Two animals remained unvaccinated. Sera were collected at 0, 28 and 60 days post vaccination (dpv) and analyzed by liquid-phase blocking ELISA (LPBE) and virus neutralization test (VNT). The experiment was conducted in accordance with international welfare guidelines and approved by the Institutional Committee for Care and Use of Experimental Animals, CICUAE-CICVyA, INTA, Argentina (CICUAE-INTA 26-2019).

### 2.7. Detection of Anti-FMDV Antibodies by Liquid-Phase Blocking ELISA (LPBE)

Total antibodies against A/Arg/2001 (A2001) FMDV strain were measured using a liquid-phase blocking ELISA (LPBE) originally developed by Hamblin et al. [20] and further modified by Periolo et al. [21].

### 2.8. Detection of Anti-FMDV Antibodies by Virus Neutralization Test (VNT)

Neutralizing antibody titers of sera were determined against A2001 Argentina strain using virus neutralization tests (VNT) according to the protocol outlined in the WOAH Manual [19]. The titers were calculated as the antibody dilution required to neutralize 50% of virus/cell mixtures at a virus dose of 100 Tissue Culture Infective Dose 50 (TCID50) according to the Spearman–Kärber method [22] and presented as log10 of the reciprocal serum dilution.

## 3. Results

### 3.1. Optimization of the High-CHO-Cell-Density Transient Gene Expression System for FMDV VLPs

#### 3.1.1. Key Features of FMDV VLPs Expression

Although CHO cells are the most commonly used cell lines for approved biopharmaceuticals such as antibodies, this is the first report using CHO cells for FMDV VLPs expression for vaccine purposes. Thus, an optimization of the transfection conditions was required to achieve a high-level expression system, considering FMDV VLPs’ key features. FMDV VLPs are expressed intracellularly, and their formation requires the expression of both P12A polyprotein and protease 3C. Once the polyprotein is expressed, the protease cleaves it, releasing VP0, VP1 and VP3, which are then assembled to form protomers, pentamers and finally the VLPs. Both proteins are toxic for mammalian cells, especially protease 3C, as it cleaves host cell proteins, including critical cellular factors. We previously attempted to produce VLPs by expressing VP0, VP1 and VP3 as separate proteins, but this approach was unsuccessful. Thus, for FMDV VLP production in CHO cells, we used the plasmid constructions detailed in Figure 1. Due to the lack of cross-protection between serotypes and potentially incomplete protection between some strains within the same serotype, vaccines should include the appropriate vaccines strains to achieve protection against circulating viruses. Therefore, demonstrating the capability of the TGE system in CHO cells to produce VLPs from various serotypes is a crucial aspect for vaccine development.

In this case, we used chimeric constructs because they possess hybrid properties of the serotypes they originate from, representing an interesting alternative for developing novel vaccines against different FMDV serotypes.

#### 3.1.2. Harvest Time

To determine the CHO-3E7 harvest time post-transfection for FMDV VLPs expression, PEI transfections were carried out using the pTT5-P12A3C plasmid. Cells were harvested at 24 and 48 hpt and cell lysates were analyzed by Western blot and ELISA (Figure 2). As shown in Figure 2, the harvest is better at time point 24 h than at 48 h (highest VP0 expression).

#### 3.1.3. Optimal Amount of P12A and 3C

As previously reported by many authors, FMDV protease 3C is toxic for mammalian cells. Thus, increasing the amount of P12A polyprotein and reducing the amount of protease 3C had been shown to increase VLPs yields in various expression systems [7,23,24]. In this direction, we optimized the ratio between both proteins by varying the amounts of pTT5-P12A3C, pTT5-P12A and pTT5-3C plasmids to be transfected. We combined either pTT5-P12A with pTT5-3C or pTT5-P12A3C with pTT5-P12A. Figure 3 shows a significant increase in the recombinant VP0/VP1protein expression levels when the amount of cDNA coding for protease 3C is reduced. Under these conditions, the intensity of the P12A band also increases, indicating that there was not enough protease 3C expressed to completely cleave the P12A polyprotein. When pTT5-P12A and pTT5-3C plasmids were combined, the highest yields were achieved with a ratio of 1000 to 1, respectively. When pTT5-P12A and pTT5-P123C plasmids were combined, the highest VP0/VP1 expression was achieved with a plasmid ratio of 99%/1%, respectively. To further adjust the ratio, additional experiments were conducted. Since the 97% P12A and 3% P12A3C ratio provided the maximum yield among all tested ratios, all subsequent experiments were performed using this condition.

#### 3.1.4. Design-of-Experiments: Optimization Using GFP as a Model Protein

We continued the optimization process, aiming to improve cDNA and PEI concentrations, by following a previously described Design of Experiments (DoE) method [25] with slight modifications. For FMDV VLPs, the transfected cultures were harvested at 24 hpt due to the toxicity of protease 3C. Taking this into account, we used GFP expression as an intracellular protein model and implemented a DoE method to achieve the highest percentage of GFP-positive cells at 24 hpt. Since CHO-3E7 transfection using BCDT media supports high cell density growth, we evaluated cell transfection at cell densities of 5 × 10^6^ and 7 × 10^6^ viable cell/mL, in both cases without dilution prior to transfection. A 2-variable central composite design with ranges for cDNA and PEI concentrations set at 1.5–3.0 μg/mL and 7.5–15.0 μg/mL, respectively, for cells at 5 × 10^6^ per mL and 2.5–7.5 μg/mL and 12.5–30.0 μg/mL, respectively, for cells at 7 × 10^6^ per mL using Reliasoft DOE++ software (https://www.hbkworld.com/en/products/software/reliability#!ref_reliasoft.com) was generated (Figure 4).

To ensure reproducibility, the center point in both designs (2.25 μg/mL DNA and 11.25 μg/mL PEI for cells at 5 × 10^6^ and 5 μg/mL DNA and 21.25 μg/mL PEI for cells at 7 × 10^6^) was repeated five times (Figure 5).

Experimentally, the highest percentage of GFP+ cells was achieved at 2.25 μg/mL cDNA and 11.25 μg/mL PEI for cells at a density of 5 × 10^6^ viable cells per mL and 2.5 μg/mL cDNA and 12.5 μg/mL PEI for cells at a density of 7 × 10^6^ viable cells per mL. However, after DoE optimization, the predicted optimal cDNA and PEI concentrations were 2.6 μg/mL and 13.4 μg/mL, respectively, for cells at densities of 5 × 10^6^ viable cells per ml, and 4.6 μg/mL and 16.1 μg/mL for cells at densities of 7 × 10^6^ viable cells per ml. Finally, to determine the cDNA and PEI transfection conditions, we compared the GFP+ cells obtained using the predicted optimal cDNA and PEI concentrations with experimentally yielding the highest transfection efficiencies.

### 3.2. Expression of VLPs of FDMV in CHO-3E7 with the Optimized Conditions

Using the optimized transfection conditions detailed in Table 1, we conducted independent expression experiments to produce FMDV VLPs from the complete A2001 Argentina strain, another serotype A strain from the ASIA topotype (A/ASIA), and two chimeric constructs carrying the VP1 protein from either the O1 Campos strain (serotype O) or the A/ASIA strain, both within the A2001 Argentina strain P12A polyprotein background (A/O and A/A, respectively). Figure 6 shows the Western Blot analysis of the cell lysates. All transfection conditions yielded similar results. Moreover, we compared the CHO cell results with our previous work [6] conducted in 293-6E cells using Western Blot and ELISA, demonstrating CHO cells give higher yields (Figure 7). Table 2 highlights the advantages and disadvantages of both methodologies in mammalian cell lines.

### 3.3. Immunogenitcity Evaluation of VLPs

#### 3.3.1. Vaccine Formulation and Swine Immunization Schedule

For vaccine formulation, CHO cell lysates were used in W/O emulsion containing 5, 25 and 50 μg of A2001 Argentina VLPs per dose. VLPs in the cell lysates were characterized by sucrose gradients, and more than 95% was found in the VLP form (Figure 8). The use of crude lysates with no further purification steps represents an advantage, considering the low prices the veterinary vaccines must have to remain competitive in the market. Figure 9 shows the immunization schedule.

#### 3.3.2. Swine Immune Response

All groups showed mean antibodies titer, determined by VNT and LPBE, above protection values (titer 1.43 for VNT and 2.20 for LPBE) at 28 days post vaccination (dpv) for FMDV A2001 Argentina strain. All single-vaccinated-groups maintained antibody titers above protection values by both methods up to 60 dpv. A booster at 28 dpv further increased antibody titers in all three groups, reaching similar levels to those elicited by the inactivated vaccine at 60 dpv (Figure 10).

## 4. Discussion

Many efforts have been made in the last decade to develop a novel vaccine against FMD. Most of these initiatives rely on the use of FMDV VLPs since the high repetitiveness of structural proteins and particulate form have been demonstrated to be most effective in developing a protective immune response in various animal models. Additionally, different expression systems and genetic strategies have been reported by many authors to produce recombinant VLPs. However, most of these approaches have only achieved publication, not commercialization. Thus far, only one technology has gained the status of novel vaccine approved for emergency purposes in the United States. This vaccine utilizes an adenovirus to express FMDV VLPs as a genetic immunization strategy. While promising for emergency use, scaling up production of this technology can be complex to meet vaccine demands in endemic regions. We have previously reported the use of transient gene expression in 293-6E cells for VLPs production and demonstrated not only immunogenicity but also scalability and reproducibility of the technology. In this report, we explored the use of CHO-3E7 cells, as they are the most widely used host cell line for the production of recombinant therapeutic proteins. Advantages of this host include the availability of multiple commercial growth media, the high cell densities that CHO cultures can reach, the favorable regulatory landscape, and the biosafety level 1 requirement for its culturing. Using this host cell line, we were able to increase the volumetric yields of recombinant VLPs up to five times compared to the 293-6E cell platform. Although the comparison between CHO and 293-6E cells was performed using different harvest times (24 h vs. 48 h, respectively), this choice was based on prior optimization studies where 293-6E cells reached maximum expression at 48 h, while CHO cells performed best at 24 h. Therefore, each system was evaluated under its own optimal conditions to reflect their realistic performance for VLP production. To support the optimization of transfection parameters, we employed a Design-of-Experiments (DoE) approach using GFP as an intracellular reporter protein. Although the highest GFP expression was observed under an experimental condition that slightly differed from the predicted optimal point, the discrepancy was minimal and within the expected range of biological variation. Such differences are commonly reported in transient gene expression systems due to assay sensitivity and cellular variability. Importantly, the best-performing experimental condition was very close to the predicted optimal range, reinforcing the value of DoE in guiding efficient and reproducible optimization while minimizing the number of experiments required. Moreover, different serotypes and chimeric constructs were successfully expressed in our CHO cell system, and recombinant A2001 Argentina strain VLPs were shown to be immunogenic in pigs.

To advance this new prototype vaccine, it is essential to conduct expression and stability studies using other VLPs serotypes, both individually and in combination. Furthermore, it is crucial to demonstrate that additional VLPs serotypes elicit protective effects in target species, in particular in cattle. In conclusion, transient gene expression in CHO cells indicate initial feasibility at small scale for the development of a recombinant vaccine against Foot-and-Mouth Disease Virus.

## 5. Conclusions

A novel high-CHO-cell-density transient gene expression system was optimized for FMDV VLP production. High yields were achieved, and VLPs from different FMDV serotypes were successfully produced. Additionally, the immunogenicity of A2001 Argentina VLPs produced using this system was demonstrated in swine.

## Figures and Tables

**Figure 1 vaccines-13-00581-f001:**
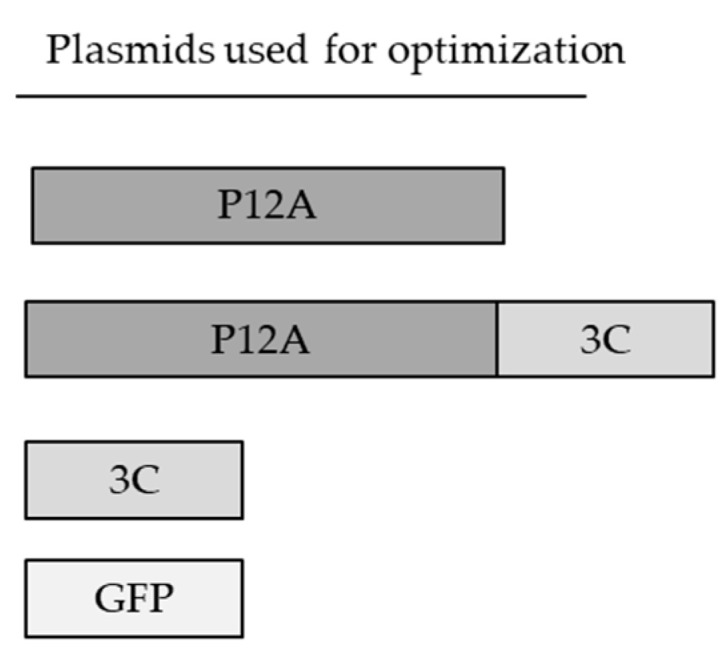
For plasmids synthesis, FMDV A2001 Argentina strain sequence was used. P12A sequences were codon-optimized for mammalian expression.

**Figure 2 vaccines-13-00581-f002:**
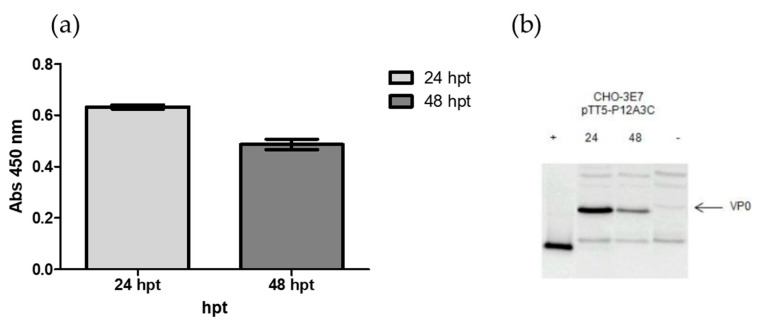
Expression levels were quantified by ELISA (**a**) and protein expression was confirmed by Western blotting (**b**). Line 1 corresponds to the VP1/VP3 proteins from A2001 virus as positive control and line 4 to untransfected CHO-3E7 cells.

**Figure 3 vaccines-13-00581-f003:**
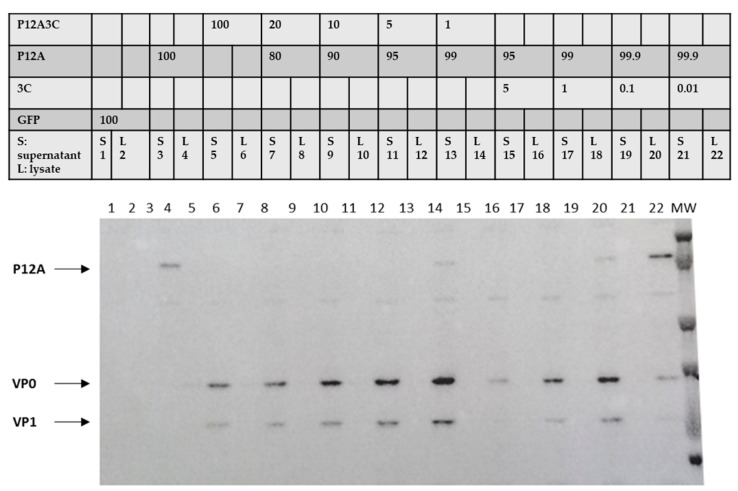
CHO-3E7 cells were transfected with different combinations of pTT5-P12A3C, pTT5-P12A and pTT5-3C plasmids, harvested 24 hpt and analyzed by Western Blot. S supernatant, L lysate. GFP was used for transfection control.

**Figure 4 vaccines-13-00581-f004:**
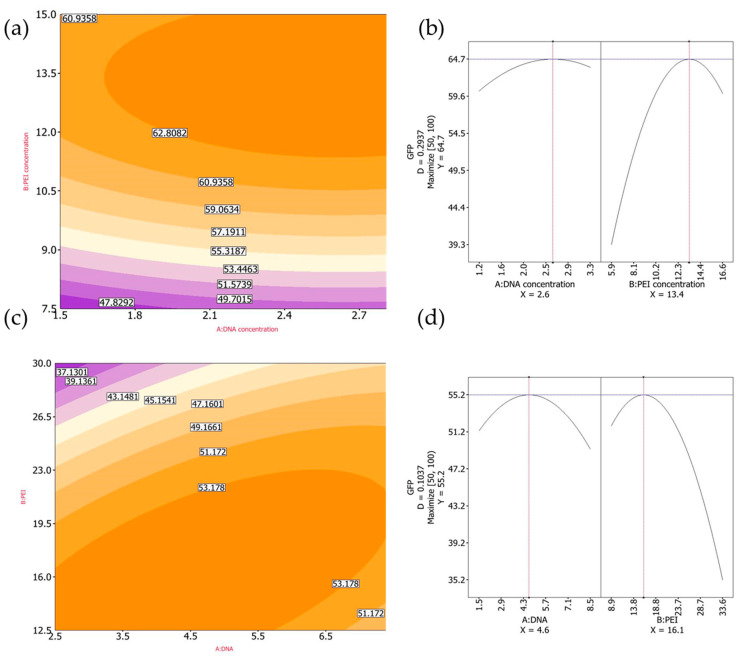
CHO-3E7 cells were transfected with different combinations of cDNA and PEI using GFP as an intracellular model protein. Cells were harvested at 24 hpt. (**a**) GFP+ cells for cells transfected at a density of 5 × 10^6^ viable cells/mL, (**b**) optimal predicted cDNA and PEI concentration when cells were transfected at a density of 5 × 10^6^ viable cells/mL, (**c**) GFP+ cells when cells were transfected at a density of 7 × 10^6^ viable cells/mL, (**d**) optimal predicted cDNA and PEI concentration when cells were transfected at a density of 7 × 10^6^ viable cells/mL.

**Figure 5 vaccines-13-00581-f005:**
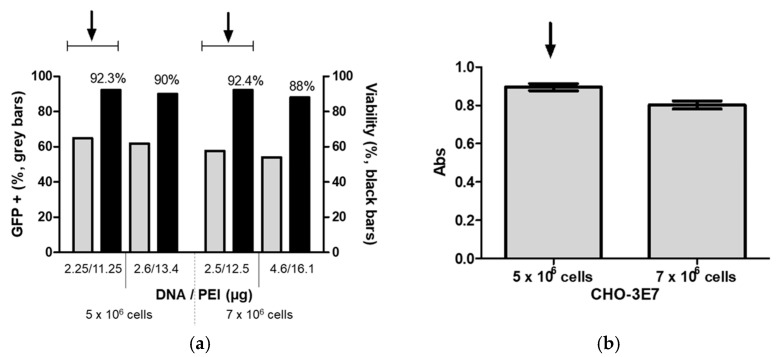
(**a**) CHO-3E7 cells were transfected with GFP at cell densities of 5 × 10^6^ viable cells/mL and 7 × 10^6^ viable cells/mL with the DNA and PEI concentrations that gave the highest GFP+ cell value experimentally and the predicted value based on the DOE results. (**b**) CHO-3E7 cells were transfected with FMDV VLPs encoding plasmids at densities of 5 × 10^6^ viable cells/mL and 7 × 10^6^ viable cells/mL at day of transfection with the DNA and PEI concentration that gave the highest GFP+ cell value experimentally and analyzed with ELISA. The experiments were repeated three times independently at figure (**a**) and five times at figure (**b**). Arrows indicate the DNA and PEI concentrations, and cell density selected for future experiments.

**Figure 6 vaccines-13-00581-f006:**
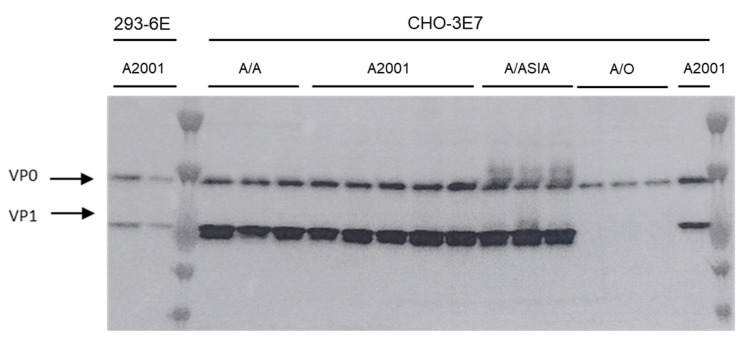
CHO-3E7 cells were transfected with the optimized transfection conditions with FMDV VLPs encoding plasmid from different serotypes and chimeras. Cells were harvested at 24 hpt and cell lysates were analyzed by Western Blot using guinea pig polyclonal antibodies anti A2001 Argentina strain. (experiments were performed in triplicate for A2001 (293-6E cells) and AA, A2001, A/Asia and AO (CHO-3E7) cells and quintuplicate for A2001 (CHO-3E7).

**Figure 7 vaccines-13-00581-f007:**
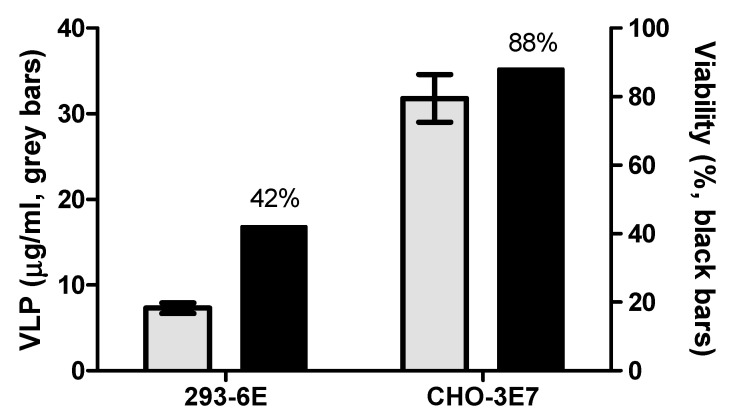
CHO-3E7 and 293-6E cells were transfected with the optimized transfection conditions with FMDV VLPs encoding plasmid and compared yield with ELISA. Also, the viability of cell culture was measured.

**Figure 8 vaccines-13-00581-f008:**
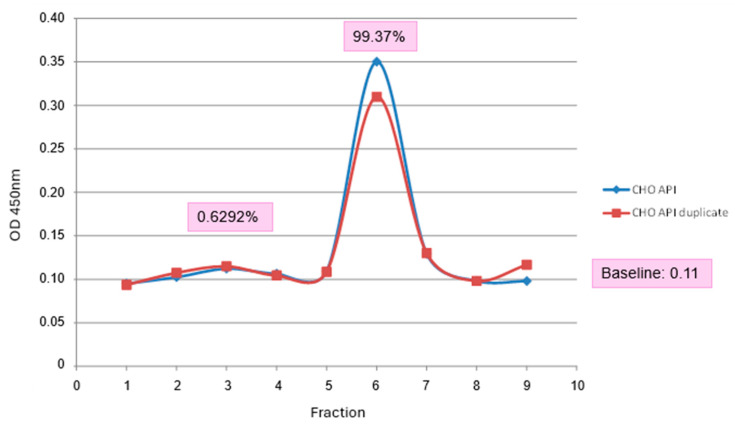
Sucrose gradient characterization of CHO cells crude lysates 24 hpt. A2001 Argentina VLPs (CHO API) were loaded onto a 45–15% sucrose gradient. Fractions were collected (*X*–axis) and analyzed by solid phase ELISA. The OD 450 (*Y*-Axis) shows the absorbance at the position of empty capsids (99.37%) and pentamers (0.6292%).

**Figure 9 vaccines-13-00581-f009:**
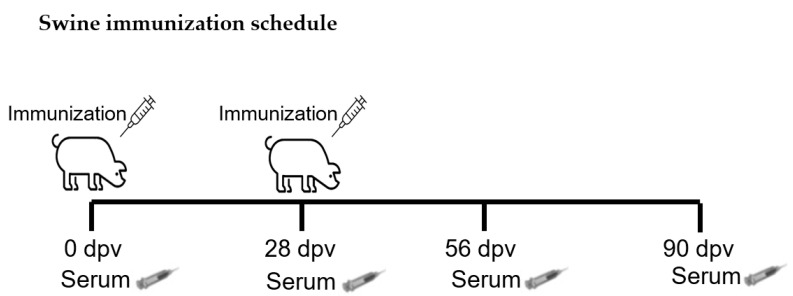
Vaccination schedule. Swine were immunized at 0 and 28 dpv with W/O vaccines containing 5 µg, 25 µg or 50 µg of A2001 Argentina VLPs or with a vaccine containing inactivated virus. Serum samples were collected at 0, 28 and 60 days post vaccination.

**Figure 10 vaccines-13-00581-f010:**
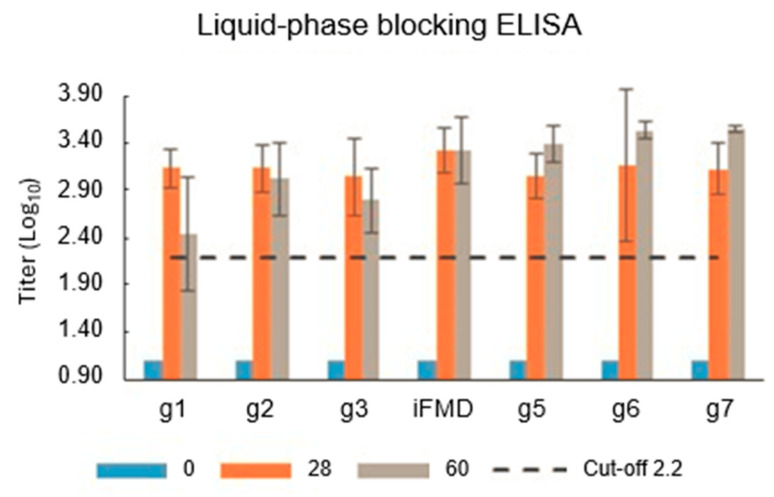
FMDV antibodies by Liquid-phase blocking ELISA. Conditions and experimental groups are described in Table 3. Cut off A2001 Argentina 2.2 log10 [26].

**Table 1 vaccines-13-00581-t001:** Optimized transfection conditions for FMDV VLP production in CHO-3E7 cells.

Optimized Condition	Value
Viable cell density at day of transfection	5 × 10^6^ cells/mL
pTT5-P12A/pTT5-P12A3C ratio (%)	97/3
Harvest time	24 hpt
DNA concentration	2.25 μg/mL
PEI concentration	11.25 μg/mL

hpt: hours post transfection.

**Table 2 vaccines-13-00581-t002:** Comparison of 293-6E and CHO-3E7 cells for FMDV VLPs production.

Title 1	293-6E	CHO-3E7
Yield	+	+++
Cell density	2 × 10^6^ cell/mL	5 × 10^6^ cell/mL
Harvest time	48	24
DNA/mL of cell culture (μg)	1	2.25
PEI/mL of cell culture (μg)	1.25	11.25
Optimized plasmid ratio	99% P12A + 1% 3C	97% P12A + 3% P12A3C
Availability of medium	+	+++
Biosecurity Level	II	I
Regulatory Concerns	+++	+

**Table 3 vaccines-13-00581-t003:** Virus neutralization test (Log_10_).

Group Dose	Immunization	28 dpv	60 dpv
G1 5 ug	0 dpv	1.63 ± 0.43	1.78 ± 0.38
G2 25 ug	0 dpv	1.56 ± 0.2	1.93 ± 0.25
G3 50 ug	0 dpv	1.49 ± 0.17	1.72 ± 0.15
iFMDV	0 dpv	2.38 ± 0.25	2.65 ± 0.49
G5 5 ug	0 and 28 dpv	1.62 ± 0.23	2.45 ± 0.29
G6 25 ug	0 and 28 dpv	1.86 ± 0.33	2.71 ± 0.3
G7 50 ug	0 and 28 dpv	1.76 ± 0.23	2.71 ± 0.26

Cut off VNT A2001 Argentina: 1.43 log10 [27].

## Data Availability

The original contributions presented in this study are included in the article. Further inquiries can be directed to the corresponding author.

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
