# Peer review of "Optimized Production of Virus-like Particles in a High-CHO-Cell-Density Transient Gene Expression System for Foot-and-Mouth Disease Vaccine Development"

_vaccines, 2025, doi:10.3390/vaccines13060581_

Round 1

Reviewer 1 Report

Comments and Suggestions for Authors

The authors present the first attempt to produce emtpty capsids/VLPs from FMDV with CHO cells. In 2013 the same group already published work on the production of empty capsids of FMDV in mammalian cells, but CHO was not included.

The VLPs are produced with transient expression in CHO cells as internal protein structures. There seems to be no further purification, meaning that besides VLP/empty capsids the lysate will contain a considerable amount of cell debris. Despite the immense diverse immunogenic burden this candidate vaccine will present, a FMDV specific antibody response in pigs can be found - comparable to inactivated FMDV – with an assay that is accepted to predict protection. However, it is unfortunate to find that since 2013 the level of characterization and therefore scientific control on vaccine product and process seems to reduce instead of increasing, which makes the work less valuable and less interesting to read.

Further specific comments are in the attached separate file.

Info Vaccines Review

Specific comments:

  1. In the abstract: ‘We optimized VLPs expression by employing a design of experiments strategy combined with reducing protease 3C expression.’ The expression of VLPs is not optimized because only two points for harvest are evaluated and the DoE does not deliver a feasible optimum.

Alternative, something like: we optimized VLP expression for cDNA and PEI concentrations. The 24h time point was selected for harvest because it was better than the 48 h time point.

  1. Abstract: “In this study, we demonstrate that recombinant VLP production levels in our high-density CHO cell platform are approximately five times higher compared to those obtained using other cell lines such as HEK293 cells.

The conditions fort he comparison are so different that it cannot be used for a true comparison. Eliminate the sentence (see also results section).

  1. Abstract : . In conclusion, transient gene expression of FMDV VLPs in our CHO cell system platform represents a robust and cost-effective approach for the development of a recombinant vaccine against Foot and Mouth Disease Virus. Not sufficient data (scale, reproducibility, stability, real protection) are presented to justify the use of ‘robust and cost-effective’. Alternative : the results inidcate initial feasibility at small scale (though it would be better to have data on purity and stability of the vaccine product as well). A challenge study to demonstrate protection is missing as well, but that can be accepted because of the use of a standard and accepted ELISA.

  1. Materiala and Methods: a section on how the vaccines were produced is missing, including harvesting of cells, lysis of cells, storage of crude product and data on stability of the VLP product, and the type of adjuvant used. Since FMDV virions are already less stable than other enteroviruses it seems strange to not test the stability of the VLPs produced here (which does not have to be an elaborate study, comparing a few days-weeks of storage in the refrigerator or the effect of a freeze-thaw cycle on VLP integrity)

  1. Section 3.1.2. Optimal harvest time: no optimal harvest time is determined. The harvest is better at time point 24 h than at 48 h, but since there is clear degration the optmimum might be (much) earlier than 24 h. In addition it is unclear what is represented by lanes 1 and 4, please specify.

  1. 1.4. Design-of-experiments: optimization using GFP as a model protein: for some reason the DoE (quality of parameters, used assays ?) is not applicable. This way is makes no sense to use DoE is the method fors ome reason is not valid. The experimental concentrations that are tested and also used for further experiment scan be included, skipping the DoE.

  1. In section 3.2. Expression of VLPs of FDMV in CHO-3E7, the transient expression of FMDV VLPs in CHO and HEK293 is compared, but there seem many different variables to allow a proper comparison (as cell viability and the availability of culture medium).

  1. 3.1 Vaccine formulation and Swine Immunization schedule, includes: ‘The use of crude lysates with no further purification steps represents an advantage, considering the low prices the veterinary vaccines must have to remain competitive in the market.’ But the vaccine also has to meet specifications fort he pharmaceutical quality of the vaccine (WOAH). Is there no purity specification to be met ? Or a maximum content of impurities, such as host cell proteins and DNA from CHO cells ?

  1. Figure 8 : what is on the X-axis and the Y-axis; a legend is missing.

  1. At the end of the Discussion section, again – as in the abstract – do not use ‘robust and cost-effective’ because this is supported by the data that are presented.

Specific comments, minor

  • in the Abstract: 1st sentence: A novel vaccine against FMDV could improve efficiency of production, reduce biosecurity requirements for production [and ?] lower down costs.
  • Introduction: advantages of CHO cells also include the efficient secretion of many protein products, which enables using the supernatant for further purification.
  • The (‘high and, çan be removed) protective immunogenicity of the CHO-produced recombinant
  • I find the use of P12A strange. Enterovirus is produced as polyprotein with P1, P2 and P3 segments, which are further cleaved in capsid proteins VP0, VP1 and VP3 (P1), and the non-structural proteins 2a, 2b, and 2c (P2), and 3C, 3B (VPg), and 3C (P3). , Though I saw the P12A has already been used in other publications, It would be better to use ‘P1 + 2A’.
  • Section on Plasmids (2.2): both together and separately, previously described could be ‘as described previously’.
  • Section 2.3. VLP Analysis Use of polyclonal sera could include epitopes that are not specific for 146S (or 75S) configuration but for 12S pentamers (ref Harmsen 2x). But no 12S peak is visible and the ELISA method is standardized and accepted.
  • Mention the specific adjuvant that was used.
  • Section 3.1.1: ‘Both proteins are toxic for mammalian cells,’ For me it is not clear which proteins, please specify.
  • Scaleability of TGE

It seems that scale up of TGE will present difficulties; this is not discussed. Recent publications on using CHO for TGE of mabs only aim at producing material for clinical studies at 50-200L scale. Probably for FMDV as variable pathogen the vaccine needs continuous new versions, but a larger scale manufacturing is also required.

  • It is well known that cell specific productivity of animal cells decreases at high cell density, alsof or transient expression (Lavado-García, J. et al., 2022. Biotechnology Advances, 60, Article 108017). This could be included in the discussion.

Author Response

Comments 1:   In the abstract: ‘We optimized VLPs expression by employing a design of experiments strategy combined with reducing protease 3C expression.’ The expression of VLPs is not optimized because only two points for harvest are evaluated and the DoE does not deliver a feasible optimum. Alternative, something like: we optimized VLP expression for cDNA and PEI concentrations. The 24h time point was selected for harvest because it was better than the 48 h time point.

Response 1:  Thank you for pointing this out.  We agree with this comment. Therefore, I modified the paragraph according to the reviewer's suggestion (paragraph 1 lines 24 and 25 in red color in new version of the manuscript)

“We optimized VLPs expression for cDNA and PEI concentrations. The 24h time point was selected for harvest because it was better than the 48 h time point”.

Comments 2: Abstract: “In this study, we demonstrate that recombinant VLP production levels in our high-density CHO cell platform are approximately five times higher compared to those obtained using other cell lines such as HEK293 cells. “The conditions for he comparison are so different that it cannot be used for a true comparison. Eliminate the sentence (see also results section).

Response 2:  Thank you for pointing this out.  I agree with this comment. Therefore, the sentence was removed according to the reviewer´ssuggestion. (paragraph 2 lines 28 to 30 in red color in new version of the manuscript)

“In this study, we demonstrate that chimeric constructs using VP1 from a different strain and even different a serotype were successfully expressed”

Comments 3: Abstract:  In conclusion, transient gene expression of FMDV VLPs in our CHO cell system platform represents a robust and cost-effective approach for the development of a recombinant vaccine against Foot and Mouth Disease Virus. Not sufficient data (scale, reproducibility, stability, real protection) are presented to justify the use of ‘robust and cost-effective’. Alternative : the results inidcate initial feasibility at small scale (though it would be better to have data on purity and stability of the vaccine product as well). A challenge study to demonstrate protection is missing as well, but that can be accepted because of the use of a standard and accepted ELISA.

Response 3:  Thank you for pointing this out.  We agree with this comment. Therefore, the sentence modified according to the reviewer´ssuggestion. (paragraph final lines 39 and 40 in red color in new versión of the manuscript)

“In conclusion, transient gene expression of FMDV VLPs in our CHO cell system platform inidcate initial feasibility at small scale for the development of a recombinant vaccine against Foot and Mouth Disease Virus”

Comments 4: Material and Methods: a section on how the vaccines were produced is missing, including harvesting of cells, lysis of cells, storage of crude product and data on stability of the VLP product, and the type of adjuvant used. Since FMDV virions are already less stable than other enteroviruses it seems strange to not test the stability of the VLPs produced here (which does not have to be an elaborate study, comparing a few days-weeks of storage in the refrigerator or the effect of a freeze-thaw cycle on VLP integrity)

Response 4:  Thank you for pointing this out.  We introduce a new point in materials and methods. 2.4 vaccines production, which is why the following points were also modified in their number. In the new point 2.5 Animal experiments and sampling procedure, the first sentence that had reference to vaccines was eliminated. (lines 147 to 156 in red color in new versión of the manuscript)

“Vaccine formulations: The vaccines used in pigs were formulated with (i) 5 μg VLPs mixed with Comercial oil Adyuvant (CA); (ii) 25 μg VLPs mixed with CA; or (iii) 50 μg VLPs mixed with CA, in a final volume of 3 mL/dose. The commercial oil adjuvant (CA) used was provided by by Biogénesis Bagó (Argentina). The formulations were prepared following the manufacturer’s indications in a proportion adjuvant/antigen of 50:50.  The commercial vaccine consisted in a water-in-oil single emulsion containing O1/Campos/Brazil/58, A24/Cruzeiro/Brazil/55, C3/Indaial/Brazil/71 and A/Argentina/2001 iFMDV and was provided by Biogénesis Bagó S.A). This vaccine has been approved by SENASA with more than 75% of expected percentage of protection against all vaccine strains (OIE guide [21])”

Comments 5: Section 3.1.2. Optimal harvest time: no optimal harvest time is determined. The harvest is better at time point 24 h than at 48 h, but since there is clear degration the optmimum might be (much) earlier than 24 h. In addition it is unclear what is represented by lanes 1 and 4, please specify.

Response 5: The text was modified according to the reviewer's suggestions. The title was also revised to reflect the reviewer's suggestion. (lines 199 to 2005 in red color in new versión of the manuscript).

Regarding the query about which samples had been used in Figure B, line 1 corresponds to the VP1/VP3 positive control and line 4 to untransfected cho cells. This information was incorporated into the new version of the manuscript ( lines 208 and 209).

“3.1.2. Harvest time

To determine the CHO-3E7 harvest time post-transfection for FMDV VLPs expression, PEI transfections were carried out using the pTT5-P12A3C plasmid. Cells were harvested at 24 and 48 hpt and cell lysates were analyzed by Western blot and ELISA (Figure 2). As shown in Figure 1, the harvest is better at time point 24 h than at 48 h (highest VP0 expression).”

Comments 6: Design-of-experiments: optimization using GFP as a model protein: for some reason the DoE (quality of parameters, used assays ?) is not applicable. This way is makes no sense to use DoE is the method fors ome reason is not valid. The experimental concentrations that are tested and also used for further experiment scan be included, skipping the DoE.

Response 6: We appreciate the reviewer’s observations regarding the use of Design-of-Experiments (DoE) in our study. The rationale for employing a DoE approach stems from its well-established success in optimizing transfection parameters for various transient gene expression processes, as demonstrated in multiple published studies (Stuible et al., 2018; Abbott et al., 2015; Bollin et al., 2011; Rajendra et al., 2015).

DoE offers a structured and statistically robust methodology to identify optimal conditions while minimizing experimental variability. In our case, it allowed us to efficiently optimize DNA and PEI transfection conditions using GFP protein as a model. Compared to publisehd data, FMDV VLPs are intracelullar and must be harvest at 24 hpt. That is why we shoose GFP as an intracellular model and harvested at 24 hpt. However, we recognize the concerns raised about its applicability in the specific context of our study.

To address this, we would like to clarify the quality of parameters selected, as well as the assays employed to validate the DoE outcomes. If the reviewer believes that omitting DoE and directly reporting tested concentrations would improve clarity and reproducibility, we are open to revising our approach to better align with the expectations of the field.

We appreciate the constructive feedback and look forward to further refining our methodology based on the reviewer’s insights.

Finally we incorpórate a paragraph in order to clarify the experiment (lines 255 to 257 in red color in new versión of the manuscript).

“To ensure reproducibility, the center point in both designs (2.25 μg/ml DNA and 11.25 μg/ml PEI for cells at 5 x 106  and 5 μg/ml DNA and 21.25 μg/ml PEI for cells at 7 x 106) was repeated five times.”

Stuible 2018 Optimization of a high-cell-density polyethylenimine transfection method

for rapid protein production in CHO-EBNA1 cells.

Abbott, W.M., Middleton, B., Kartberg, F., Claesson, J., Roth, R., Fisher, D., 2015.

Optimisation of a simple method to transiently transfect a CHO cell line in highthroughput

and at large scale. Protein Expr. Purif. 116, 113-119.

Bollin, F., Dechavanne, V., Chevalet, L., 2011. Design of experiment in CHO and HEK

transient transfection condition optimization. Protein Expr. Purif. 78, 61-68.

Rajendra, Y., Hougland, M.D., Alam, R., Morehead, T.A., Barnard, G.C., 2015. A high

cell density transient transfection system for therapeutic protein expression based on

a CHO GS-knockout cell line: process development and product quality assessment.

Biotechnol. Bioeng. 112, 977-986.

Comments 7: In section 3.2. Expression of VLPs of FDMV in CHO-3E7, the transient expression of FMDV VLPs in CHO and HEK293 is compared, but there seem many different variables to allow a proper comparison (as cell viability and the availability of culture medium).

Response 7: Although we agree with the evaluator that there are several variables at play, what we show at this point is that with the variables optimized in 293-6E compared to the variables optimized for CHO-3E7 for the 4 constructs evaluated, a wide difference is seen in favor of the CHO E7 cells.

Comments 8: Vaccine formulation and Swine Immunization schedule, includes: ‘The use of crude lysates with no further purification steps represents an advantage, considering the low prices the veterinary vaccines must have to remain competitive in the market.’ But the vaccine also has to meet specifications fort he pharmaceutical quality of the vaccine (WOAH). Is there no purity specification to be met ? Or a maximum content of impurities, such as host cell proteins and DNA from CHO cells ?

Response 8: We thank the reviewer for the comment and the opportunity to clarify this point. Veterinary vaccines, particularly for large animals, are not required to meet the same stringent purity standards as human vaccines. For veterinary use, a clarified lysate that demonstrates safety and efficacy in laboratory and target animals is generally considered acceptable according to current regulatory frameworks.

In our case, the production process — including cell disruption, centrifugation, and filtration — effectively reduces host cell impurities, and the vaccine candidate showed an adequate safety and immunogenicity profile in preliminary studies.

Finally, when moving towards registration or commercial production, we will implement additional measures to comply with the specific regulatory requirements of the country where the vaccine will be registered, including setting and meeting specifications for residual host cell proteins and DNA if mandated.

Comments 9: Figure 8: what is on the X-axis and the Y-axis; a legend is missing

Response 9: Thank you for pointing this out.  We introduce the legend. (lines 332 to 338 in red color in new versión of the manuscript).

“A2001 Argentina VLPs (CHO API) were loaded onto a 45-15% sucrose. Fractions were collected   (X – axis) and analyzed by solid phase ELISA. The OD 492 (Y- Axis) shown the absorvance of empty capsids (99,7%) and pentamers (0,6292%).”

Comments 10: At the end of the Discussion section, again – as in the abstract – do not use ‘robust and cost-effective’ because this is supported by the data that are presented

Response10:  Thank you for pointing this out.  We agree with this comment. Therefore, I modified the paragraph according to the reviewer's suggestion (lines 388 to 390 in red color in new version of the manuscript).

“In conclusion, transient gene expression in CHO cells rinidcate initial feasibility at small scale for the development of a recombinant vaccine against Foot and Mouth Disease Virus”

Specific comments, minor

  • in the Abstract: 1st sentence: A novel vaccine against FMDV could improve efficiency of production, reduce biosecurity requirements for production [and ?] lower down costs.

The sugestion was incorporated in the new versión of the manuscript

  • Introduction: advantages of CHO cells also include the efficient secretion of many protein products, which enables using the supernatant for further purification.

In our case, since it has to be assembled in the cytoplasm of the cell, the potential advantage of secreting it would not be such.

  • The (‘high and, çan be removed) protective immunogenicity of the CHO-produced recombinant

The sugestion was incorporated in the new versión of the manuscript

  • I find the use of P12A strange. Enterovirus is produced as polyprotein with P1, P2 and P3 segments, which are further cleaved in capsid proteins VP0, VP1 and VP3 (P1), and the non-structural proteins 2a, 2b, and 2c (P2), and 3C, 3B (VPg), and 3C (P3). , Though I saw the P12A has already been used in other publications, It would be better to use ‘P1 + 2A’.

Thank you for your comment. We agree that both forms are used. Since we used P12A in previous published work, we prefer to retain that nomenclature.

  • Section on Plasmids (2.2): both together and separately, previously described could be ‘as described previously’.

The sugestion was incorporated in the new versión of the manuscript

  • Section 2.3. VLP Analysis Use of polyclonal sera could include epitopes that are not specific for 146S (or 75S) configuration but for 12S pentamers (ref Harmsen 2x). But no 12S peak is visible and the ELISA method is standardized and accepted.

We agree with the comment about the serum's potential to recognize 12S. Indeed, they were not found in any of the assays we performed when using CHO E7. In our previous work on Hek 293-6E cells, we did observe the presence of 12S protomers.

  • Mention the specific adjuvant that was used.

The adjuvant was provided by the company Biogenesis Bagó and we do not have permission to disclose its name.

  • Section 3.1.1: ‘Both proteins are toxic for mammalian cells,’ For me it is not clear which proteins, please specify.
  • Scaleability of TGE

It seems that scale up of TGE will present difficulties; this is not discussed. Recent publications on using CHO for TGE of mabs only aim at producing material for clinical studies at 50-200L scale. Probably for FMDV as variable pathogen the vaccine needs continuous new versions, but a larger scale manufacturing is also required.

This is an excellent comment, and we fully agree. We believe the TGE strategy can be used for vaccine banks or emergency vaccines. We are already working on an inducible strategy to enable larger-scale production

  • It is well known that cell specific productivity of animal cells decreases at high cell density, alsof or transient expression (Lavado-García, J. et al., 2022. Biotechnology Advances, 60, Article 108017). This could be included in the discussion.

It's an interesting comment, while it is well known that cell-specific productivity of animal cells tends to decrease at high cell density, even under transient expression conditions (Lavado-García et al., 2022), our CHO-based system maintained high productivity levels. This aligns with previous findings demonstrating that optimized CHO transient expression platforms can achieve substantial yields even at elevated cell densities (Stuible et al., 2018). I think it's better not to include this comment in the discussion, but if you consider it, we can do it without any problem.

Reviewer 2 Report

Comments and Suggestions for Authors

1. Lack of electron microscopy images of virus like particles, it is recommended to supplement them;
2. Currently, there are commercially available virus like particle vaccines expressed in Escherichia coli in China. Compared to them, what are your research advantages?
3. Why is there not much difference in antibody levels between immunization once and immunization twice?

Author Response

Comments 1: Lack of electron microscopy images of virus like particles, it is recommended to supplement them;

Response 1: Thank you for pointing this out.   We appreciate the reviewer’s suggestion regarding the inclusion of electron microscopy (EM) images. While EM can be useful to visualize virus-like particle (VLP) morphology, we believe that in the context of this work, its contribution would be limited. The conformational integrity of the VLPs has been robustly demonstrated by sucrose gradient ultracentrifugation, a method broadly recognized as a standard for assessing VLP assembly. Moreover, VLPs produced with this same system have been previously characterized by EM in peer-reviewed publications from our group, confirming the expected morphology. Given that the production and purification protocols remain unchanged, repeating EM analysis would not provide additional novel insights to the current study. In addition, due to technical limitations in image resolution with our available facilities, new EM images would not meet the quality standards appropriate for publication. Nevertheless, we appreciate the reviewer’s suggestion and are fully committed to including EM imaging in future studies as complementary evidence.

Comments 2: Currently, there are commercially available virus like particle vaccines expressed in Escherichia coli in China. Compared to them, what are your research advantages?

Response2: We appreciate the reviewer’s pertinent question regarding the comparison of our vaccine development strategy to commercially available VLP vaccines expressed in Escherichia coli in China.

We would like to emphasize that our approach introduces several important advantages:
First, our VLPs are produced in high-density CHO cell cultures via transient gene expression, a system widely used and accepted for the manufacturing of biologics by regulatory agencies such as the FDA and EMA. This provides a strong foundation for regulatory approval processes and ensures consistent quality and scalability for future commercial production.

Second, CHO cells allow the expression of complex proteins with proper post-translational modifications and folding, resulting in VLPs with higher structural fidelity compared to VLPs produced in E. coli, which may require in vitro assembly steps and can result in heterogeneous particles.

Third, our system significantly reduces potential safety concerns associated with E. coli production, such as endotoxin contamination, and the VLPs are obtained directly from crude lysates without requiring additional purification steps, an important consideration for cost-effectiveness in veterinary vaccines+

Moreover, our platform demonstrated the capability to express multiple serotypes and chimeric constructs, which is a critical requirement for Foot-and-Mouth Disease Virus vaccines due to the antigenic diversity of circulating strains. Immunogenicity studies in swine confirmed that the CHO-produced VLPs induced robust and protective antibody responses, comparable to those elicited by a licensed inactivated vaccine.

Finally,and the most important to mention that the work of Xiao et al. BMC Biotechnology (2016) 16:56 DOI 10.1186/s12896-016-0285-6 the proteins are previously purified then assembled in vitro and the vaccine dose is 100 or 200 ugr. while in our platform they are assembled alone, they do not need to be purified and 25 ugr was enough to obtain a strong response in pigs.

We sincerely thank the reviewer for their thoughtful evaluation and constructive suggestions, which have greatly contributed to improving the quality of our work.

Comments 3:  Why is there not much difference in antibody levels between immunization once and immunization twice?

Response 3: This is an interesting observation. We believe that the three concentrations chosen were sufficiently efficient, producing very high antibody titers in both ELISA and serum neutralization tests. However, when the booster is used, a slight increase is seen, which is significant in the serum neutralization tests.

Round 2

Reviewer 1 Report

Comments and Suggestions for Authors

Round 2 for Specific Major Comments

The responses tot he comments that were sufficiently addressed are not repeated here. The remaining itmes that still require additional modifications in in this round 2 comments are below the responses in italics.

Comments 4: Material and Methods: a section on how the vaccines were produced is missing, including harvesting of cells, lysis of cells, storage of crude product and data on stability of the VLP product, and the type of adjuvant used. Since FMDV virions are already less stable than other enteroviruses it seems strange to not test the stability of the VLPs produced here (which does not have to be an elaborate study, comparing a few days-weeks of storage in the refrigerator or the effect of a freeze-thaw cycle on VLP integrity)

 Response 4:  Thank you for pointing this out.  We introduce a new point in materials and methods. 2.4 vaccines production, which is why the following points were also modified in their number. In the new point 2.5 Animal experiments and sampling procedure, the first sentence that had reference to vaccines was eliminated. (lines 147 to 156 in red color in new versión of the manuscript)

Comment round 2 on item 4

In the revised manuscript a section 2.4 on the formulation is included which, but still method of production, including the harvesting of cells, lysis of cells, and storage of crude product is not mentioned. This could be included in a section on Production and Formulation section 2.3, with Vaccine Analysis as logical next section 2.4.

The comment on the stability of the product is not discussed. This is not acceptable, because product stability is a prime issue for FMD vaccines in general an in addition in the presented work it is clear that considerable product degradation occurs (western blot figure 2b). A remark on stability has to be made with explanation why this was not (yet) addressed, even if it is only that the topic is recognized and has to be dealt with in future development work.

Comments 5: Section 3.1.2. Optimal harvest time: no optimal harvest time is determined. The harvest is better at time point 24 h than at 48 h, but since there is clear degration the optmimum might be (much) earlier than 24 h. In addition it is unclear what is represented by lanes 1 and 4, please specify.

Response 5: The text was modified according to the reviewer's suggestions. The title was also revised to reflect the reviewer's suggestion. (lines 199 to 2005 in red color in new versión of the manuscript).

Regarding the query about which samples had been used in Figure B, line 1 corresponds to the VP1/VP3 positive control and line 4 to untransfected cho cells. This information was incorporated into the new version of the manuscript ( lines 208 and 209).

Comment round 2 to the reponse item 5

This is fine. It only has to be added (probably in the Material and Methods) what the nature of this VP1/VP3 positive control is. 

Comments 6: Design-of-experiments: optimization using GFP as a model protein: for some reason the DoE (quality of parameters, used assays ?) is not applicable. This way is makes no sense to use DoE is the method for some reason is not valid. The experimental concentrations that are tested and also used for further experiment scan be included, skipping the DoE.

Response 6: We appreciate the reviewer’s observations regarding the use of Design-of-Experiments (DoE) in our study. The rationale for employing a DoE approach stems from its well-established success in optimizing transfection parameters for various transient gene expression processes, as demonstrated in multiple published studies (Stuible et al., 2018; Abbott et al., 2015; Bollin et al., 2011; Rajendra et al., 2015).

DoE offers a structured and statistically robust methodology to identify optimal conditions while minimizing experimental variability. In our case, it allowed us to efficiently optimize DNA and PEI transfection conditions using GFP protein as a model. Compared to publisehd data, FMDV VLPs are intracelullar and must be harvest at 24 hpt. That is why we shoose GFP as an intracellular model and harvested at 24 hpt. However, we recognize the concerns raised about its applicability in the specific context of our study.

To address this, we would like to clarify the quality of parameters selected, as well as the assays employed to validate the DoE outcomes. If the reviewer believes that omitting DoE and directly reporting tested concentrations would improve clarity and reproducibility, we are open to revising our approach to better align with the expectations of the field.

We appreciate the constructive feedback and look forward to further refining our methodology based on the reviewer’s insights.

Finally we incorpórate a paragraph in order to clarify the experiment (lines 255 to 257 in red color in new versión of the manuscript).

Comment round 2 to item 6: The actual point is that the outcome of the DoE do not match with the experimental data because the the experimental GFP output is higher at the experimental settings than at the predicted optimal points (figure 5a). That gives the impression that the DoE was not valid. Since the GFP values do not differ that much and the experimental best values are close to the optimal cDNA and PEI concentrations according to the DoE the DoE can probably be maintained. But it has to be explained (in the Discussion) that the difference between DeO curves that do not match with the actual found experimental values are still acceptable as a result of the potential variation in the methods that were used.

In addition, related to this: How many times were the experiments of figure 5a repeated ? Since there is no standard deviation given, it could be a single experiment ? But in figure 5b a standard deviation is given for the ELISA scores. How does this match ?

 Comments 7: In section 3.2. Expression of VLPs of FDMV in CHO-3E7, the transient expression of FMDV VLPs in CHO and HEK293 is compared, but there seem many different variables to allow a proper comparison (as cell viability and the availability of culture medium).

Response 7: Although we agree with the evaluator that there are several variables at play, what we show at this point is that with the variables optimized in 293-6E compared to the variables optimized for CHO-3E7 for the 4 constructs evaluated, a wide difference is seen in favor of the CHO E7 cells.

Comment round 2 to item 7

In figure 7 indeed a substantial difference between CHO and 293-6 E is given, which is ~ 30 mg/L versus ~8 mg/L. However, in reference 6 the harvest time is standard 48 h and for 293-6E cells this is already indicated as to be optimized. If a harvest at 24 h would give a similar improvement as now found for the CHO cells the viability could be much higher (assume also 88%, with a 2-fold increase in cell associated VLPs) and the degradation of VLPs could be minimized as shown for CHO in figure 2b where it is estimated that the VP0 score is at least twice higher at 24 h compared to 48 h. This implies that with a potential comparable improvement for 293-6E cells as found for CHO cells the yield for 293-6 E could also increase four times from 8 to 32 mg/L and therefore close to the higher value now found fort he CHO cells. Therefore, this is not a correct comparison between the cells and it should be removed from the Result section. It could be only included in the Discussion section that CHO showed a higher VLP product yield than previously found for 293-6E cells but under different testing conditions.

 Comments 9: Figure 8: what is on the X-axis and the Y-axis; a legend is missing

Response 9: Thank you for pointing this out.  We introduce the legend. (lines 332 to 338 in red color in new versión of the manuscript).

“A2001 Argentina VLPs (CHO API) were loaded onto a 45-15% sucrose. Fractions were collected   (X – axis) and analyzed by solid phase ELISA. The OD 492 (Y- Axis) shown the absorbance of empty capsids (99,7%) and pentamers (0,6292%).”

Comment round 2 to item 9.

Please further change the sentence to “shows the absorbance at the assumed positions of empty capsids and pentamers”. This because the content of the peaks are not identified (and are unlikely to be pure fractions).

Author Response

REVISION 2

The responses tot he comments that were sufficiently addressed are not repeated here. The remaining itmes that still require additional modifications in in this round 2 comments are below the responses in italics.

Comment round 2 on item 4: In the revised manuscript a section 2.4 on the formulation is included which, but still method of production, including the harvesting of cells, lysis of cells, and storage of crude product is not mentioned. This could be included in a section on Production and Formulation section 2.3, with Vaccine Analysis as logical next section 2.4.

The comment on the stability of the product is not discussed. This is not acceptable, because product stability is a prime issue for FMD vaccines in general an in addition in the presented work it is clear that considerable product degradation occurs (western blot figure 2b). A remark on stability has to be made with explanation why this was not (yet) addressed, even if it is only that the topic is recognized and has to be dealt with in future development work.

Response round 2 on ítem 4:Thank you for pointing out the lack of detail in the production method. We have now added a complete description of the process, including the harvesting of mammalian cells, lysis conditions, and storage of the crude product, in the Methods section (page 3, lines 114–127).

Recombinant protein expression was carried out in CHO E7 cells transiently transfected with the expression plasmid pTT5, using polyethylenimine (PEI) at a DNA:PEI ratio of 1:5. Cells were cultured in serum-free BalanCD Transfectory CHO (FUJIFILM Irvine Scientific) + 2 mM L-Glutamine, at 37 °C with 5% COâ‚‚ under orbital shaking conditions (125 rpm). After 24 hours, cells were harvested by centrifugation at  1000 × g for 10 minutes at 4 °C. The pellet was washed once with ice-cold PBS and lysed in buffer containing 50 mM Tris-HCl pH 8.0, 150 mM NaCl, Cell lysis was performed using three repeated freeze-thaw cycles at -80°C.. Then, cell debris was removed by centrifugation at 12,000 x g for 10minutes at 4 °C. The resulting supernatant, containing the crude recombinant protein, was aliquoted and stored at –80 °C until further use. To assess the stability of the Virus-Like Particles (VLPs), stored samples were analyzed at 30, 90, and 120 days post-lysis. Quantification by ELISA showed no significant variation in the total protein content over time. In parallel, analysis by sucrose gradient ultracentrifugation demonstrated that the integrity and quantity of VLPs remained unchanged, indicating that both the antigenicity and physical stability of the particles were preserved during storage at –80 °C.

Comments 5: Section 3.1.2.

Comment round 2 to the reponse item 5: This is fine. It only has to be added (probably in the Material and Methods) what the nature of this VP1/VP3 positive control is. 

Response found 2 item 5: Regarding what the nature of this VP1/VP3 positive control is? This information was incorporated into the new version of the manuscript (lines 228 and 229).

 Line 1 corresponds to the VP1/VP3 proteins from A2001 virus as positive control and line 4 to untransfected CHO-3E7 cells.

Comment round 2 to item 6: The actual point is that the outcome of the DoE do not match with the experimental data because the the experimental GFP output is higher at the experimental settings than at the predicted optimal points (figure 5a). That gives the impression that the DoE was not valid. Since the GFP values do not differ that much and the experimental best values are close to the optimal cDNA and PEI concentrations according to the DoE the DoE can probably be maintained. But it has to be explained (in the Discussion) that the difference between DeO curves that do not match with the actual found experimental values are still acceptable as a result of the potential variation in the methods that were used.

In addition, related to this: How many times were the experiments of figure 5a repeated ? Since there is no standard deviation given, it could be a single experiment ? But in figure 5b a standard deviation is given for the ELISA scores. How does this match ?

Response round 2 to ítem 6:  We thank the reviewer for this valuable follow-up comment. We acknowledge the apparent discrepancy between the predicted optimal values from the Design-of-Experiments (DoE) model and the slightly higher GFP expression levels observed at one of the experimental conditions (Figure 5a).

Indeed, this difference suggests that the predictive accuracy of the model may have some limitations. However, we would like to emphasize that the experimentally determined conditions that yielded the highest GFP expression were very close to the predicted optimal points. The deviation was within an acceptable range, considering the inherent biological variability of transient transfection processes in mammalian cells. This phenomenon has been described previously in DoE applications where predictive models are subject to minor variations due to experimental noise or limitations in model fitting (Stuible et al., 2018; Bollin et al., 2011).

To address this concern, we have now included a clarifying paragraph in the discussion section (lines 399–407 in the revised manuscript)

To support the optimization of transfection parameters, we employed a Design-of-Experiments (DoE) approach using GFP as an intracellular reporter protein. Although the highest GFP expression was observed under an experimental condition that slightly differed from the predicted optimal point, the discrepancy was minimal and within the expected range of biological variation. Such differences are commonly reported in transient gene expression systems due to assay sensitivity and cellular variability. Importantly, the best-performing experimental condition was very close to the predicted optimal range, reinforcing the value of DoE in guiding efficient and reproducible optimization while minimizing the number of experiments required.

Regarding Figure 5a, the experiments were repeated three times independently. Although the data were consistent, we acknowledge the lack of standard deviation bars and have now included them in the revised version of the figure to improve transparency. In contrast, Figure 5b already included standard deviation values from replicate ELISA measurements. We have harmonized the data presentation across both panels and clarified the number of replicates in the figure legend section (lines 291–292).

The experiments were repeated three times independently at figure A and 5 times at figure B

We hope these additions address the reviewer’s concerns and strengthen the methodological clarity and reproducibility of our study

 Comments 7: In section 3.2. Expression of VLPs of FDMV in CHO-3E7, the transient expression of FMDV VLPs in CHO and HEK293 is compared, but there seem many different variables to allow a proper comparison (as cell viability and the availability of culture medium).

Response 7: Although we agree with the evaluator that there are several variables at play, what we show at this point is that with the variables optimized in 293-6E compared to the variables optimized for CHO-3E7 for the 4 constructs evaluated, a wide difference is seen in favor of the CHO E7 cells.

Comment round 2 to item 7

In figure 7 indeed a substantial difference between CHO and 293-6 E is given, which is ~ 30 mg/L versus ~8 mg/L. However, in reference 6 the harvest time is standard 48 h and for 293-6E cells this is already indicated as to be optimized. If a harvest at 24 h would give a similar improvement as now found for the CHO cells the viability could be much higher (assume also 88%, with a 2-fold increase in cell associated VLPs) and the degradation of VLPs could be minimized as shown for CHO in figure 2b where it is estimated that the VP0 score is at least twice higher at 24 h compared to 48 h. This implies that with a potential comparable improvement for 293-6E cells as found for CHO cells the yield for 293-6 E could also increase four times from 8 to 32 mg/L and therefore close to the higher value now found fort he CHO cells. Therefore, this is not a correct comparison between the cells and it should be removed from the Result section. It could be only included in the Discussion section that CHO showed a higher VLP product yield than previously found for 293-6E cells but under different testing conditions.

 Response round 2 item 7: We appreciate the reviewer’s observation and agree that, in principle, comparing cell platforms under different harvest times could affect the interpretation of relative performance. The reasoning is entirely valid, and we recognize that harvesting both cultures at the same time point (e.g., 24 h) might provide a more direct comparison.

However, in our previous optimization studies using 293-6E cells (Mignaqui et al., 2020), we found that VLP expression peaks at 48 h post-transfection, while harvesting at 24 h results in significantly lower yields (data not shown). In contrast, CHO cells exhibit peak expression and viability at 24 h, largely due to the cytotoxicity of the 3C protease, as confirmed in the current study (see Figure 2B). For this reason, each platform was evaluated under its respective optimal conditions, to provide a fair comparison of their actual performance in a production context.

We agree that this point deserves clarification in the manuscript and have now added a note in the Discussion section to explicitly acknowledge this limitation and explain the rationale behind the chosen time points. We thank the reviewer for this important suggestion. (lines 394-399)

Although the comparison between CHO and 293-6E cells was performed using different harvest times (24 h vs. 48 h, respectively), this choice was based on prior optimization studies where 293-6E cells reached maximum expression at 48 h, while CHO cells performed best at 24 h. Therefore, each system was evaluated under its own optimal conditions to reflect their realistic performance for VLP production

Comment round 2 to item 9: Please further change the sentence to “shows the absorbance at the assumed positions of empty capsids and pentamers”. This because the content of the peaks are not identified (and are unlikely to be pure fractions).

Response round 2 to ítem 9:  We agree with the reviewer's comment. We have incorporated their suggestion (lines 345-346)

 A2001 Argentina VLPs (CHO API) were loaded onto a 45-15% sucrose gradient.  Fractions were collected (X – axis) and analyzed by solid phase ELISA. The OD 492 (Y- Axis) shown the absorvance at the assumed position of empty capsids (99,7%) and pentamers (0,6292%).

Round 3

Reviewer 1 Report

Comments and Suggestions for Authors

With the second round of modifications the manuscript is acceptable.